| Editor's Pick | Computational Biology | Research Article

# Deep learning enabled integration of tumor microenvironment microbial profiles and host gene expressions for interpretable survival subtyping in diverse types of cancers

Haohong Zhang,[1] Xinghao Xiong,[1] Mingyue Cheng,[1] Lei Ji,[2,3] Kang Ning[1]

**ABSTRACT** The tumor microbiome, a complex community of microbes found in tumors, has been found to be linked to cancer development, progression, and treatment outcome. However, it remains a bottleneck in distangling the relationship between the tumor microbiome and host gene expressions in tumor microenvironment, as well as their concert effects on patient survival. In this study, we aimed to decode this complex relationship by developing ASD-cancer (autoencoder-based subtypes detector for cancer), a semi-supervised deep learning framework that could extract survival-related features from tumor microbiome and transcriptome data, and identify patients' survival subtypes. By using tissue samples from The Cancer Genome Atlas database, we identified two statistically distinct survival subtypes across all 20 types of cancer Our framework provided improved risk stratification (e.g., for liver hepatocellular carcinoma, [LIHC], log-rank test, $P = 8.12E{-}6$) compared to PCA (e.g., for LIHC, log-rank test, $P = 0.87$), predicted survival subtypes accurately, and identified biomarkers for survival subtypes. Additionally, we identified potential interactions between microbes and host genes that may play roles in survival. For instance, in LIHC, *Arcobacter*, *Methylocella*, and *Isopteri-cola* may regulate host survival through interactions with host genes enriched in the HIF-1 signaling pathway, indicating these species as potential therapy targets. Further experiments on validation data sets have also supported these patterns. Collectively, ASD-cancer has enabled accurate survival subtyping and biomarker discovery, which could facilitate personalized treatment for broad-spectrum types of cancers.

**IMPORTANCE** Unraveling the intricate relationship between the tumor microbiome, host gene expressions, and their collective impact on cancer outcomes is paramount for advancing personalized treatment strategies. Our study introduces ASD-cancer, a cutting-edge autoencoder-based subtype detector. ASD-cancer decodes the complexities within the tumor microenvironment, successfully identifying distinct survival subtypes across 20 cancer types. Its superior risk stratification, demonstrated by significant improvements over traditional methods like principal component analysis, holds promise for refining patient prognosis. Accurate survival subtype predictions, biomarker discovery, and insights into microbe-host gene interactions elevate ASD-cancer as a powerful tool for advancing precision medicine. These findings not only contribute to a deeper understanding of the tumor microenvironment but also open avenues for personalized interventions across diverse cancer types, underscoring the transformative potential of ASD-cancer in shaping the future of cancer care.

**KEYWORDS** tumor microbiome, cancer prognosis, survival subtype, deep learning

Address correspondence to Kang Ning, ningkang@hust.edu.cn.

Haohong Zhang and Xinghao Xiong contributed equally to this article. Haohong Zhang contributed to the design of the work and thus is placed before Xinghao Xiong.

The authors declare no conflict of interest.

See the funding table on p. 17.

Cancer is characterized by heterogeneous histopathological, genomic, and transcriptomic profiles within both the tumor and its microenvironment, which contribute

to variations in response rates to therapy and patient outcomes (1). The current clinical approaches to many types of cancer entail manual histopathological assessment, where tumor invasion, anaplasia, necrosis, and mitoses are used for grading and staging patients to guide therapeutic decision-making (2). However, the subjective interpretation of histopathologic features has been demonstrated to suffer from substantial inter- and intraobserver variability, resulting in varying outcomes for patients with the same grade or stage. Therefore, identification of cancer subtypes is essential for diagnosis and prognosis in clinics. Cancer subtypes are crucial as they provide patients with opportunities for personalized treatment strategies (3). Cancer subtyping is particularly useful for subtypes with similar molecular and pathway alterations because it enables the application of similar treatment modalities. Survival stratified patient subtypes represent an example of a cancer subtype with prognostic significance, and provide valuable insights into the molecular factors associated with survival (4).

Traditionally, cancer has been regarded as a disease originating from alterations in the genetic makeup of human beings (5, 6). As such, there is a long history of linking tumor gene expression to cancer outcomes (7–10). However, cancer is a complex disease that involves not only the host but also the tumor microenvironment (TME), a complex ecosystem that surrounds and interacts with cancer cells. Moreover, gene expression data has subject-specific limitations and noise (11). Therefore, focusing solely on host gene expression may neglect other molecular mechanisms within the TME. Moreover, tumors are evolutionary systems subject to natural selection operating on their genomes, enabling adaptation to the TME. Consequently, the tumor microenvironment's composition, including microbial profiles and host gene expressions, could profoundly influence the selective forces that shape the tumor genome, potentially giving rise to distinct molecular subtypes (12). Therefore, integrating multi-omics data has the potential to augment our comprehension of cancer (13, 14) and paves the way for precision medicine, which offers individualized diagnosis, prognosis, treatment, and care (15, 16).

The tumor microbiome is a complex and diverse community of microbes that inhabit human tumors and adjacent tissue (17). Poore et al. (18) recently developed a computational workflow that utilizes two orthogonal microbial detection pipelines to obtain high-quality microbial abundances from high-throughput sequencing data of human tumors. The tumor microbiome has been shown to play a function in the development (19), progression (20), and response to treatment (21) of various types of cancer. The relationship between the tumor microbiome and patient survival is the subject of ongoing research, as some studies (22, 23) have suggested that the microbiome of a tumor may impact patient survival. *Malassezia globose*, for instance, has been linked to an increased risk of death in breast cancer patients (23). However, the mechanisms by which microbiota contribute to shaping the molecular properties of the tumor and influencing clinical outcomes remain poorly understood (24–26). The integration of tumor microbiomes and transcriptomes can provide a deeper comprehension of the interactions between microbes and host genes (27), thereby improving patient prognosis by enabling clinicians to devise tailored treatment strategies based on the unique characteristics of each patient. Nevertheless, a lack of knowledge regarding the interaction between host genes and the tumor microbiome has precluded us from using them for accurate patient survival analysis.

In this study, we employed a deep learning strategy to integrate host gene expression and the tumor microbiome within the TME for cancer survival analysis. We developed a deep learning-based framework called ASD-cancer (autoencoder-based subtypes detector for cancer), which is a semi-supervised deep learning framework based on autoencoder for the detection of cancer survival subtypes. We applied this framework to RNA-seq and tumor microbiome data from 20 types of cancer from The Cancer Genome Atlas (TCGA) (28) database and identified survival subtypes with high quality for patient risk stratification. Compared to the conventional principal component analysis (PCA) technique, the ASD-cancer framework demonstrated superior performance. In

addition, we analyzed the distribution of clinical stages in our survival subtypes and discovered that several cancers have similar stage distribution patterns in both subtypes. We also found that important biomarkers for classifying survival subtypes are likely not sensitive to clinical stages, emphasizing these intricate stage-independent drivers for survival subtyping. Furthermore, our analysis revealed that the high-risk group had more cancer-related pathways compared to the low-risk group, and we identified potential survival-related pathways of interaction between microbes and host genes. Finally, we validated our framework on two external cohorts utilizing transfer learning strategy, and also obtained a clear cut in survival subtyping by using microbial features.

## RESULTS

### Ensemble deep learning-based survival subtype detection model using multi-omics data

We concentrated on 20 distinct types of cancer that are represented in the TCGA database. These types of cancer affect a wide range of organs and tissues throughout the body, and 15 of them have clinical stage information defined by the AJCC Cancer Staging System (2) (Fig. 1a; Table S1). To acquire the survival subtypes, we obtained paired gene expression and the tumor microbiome data, with gene expression data downloaded from the TCGA databases, along with the tumor microbiome data derived from the RNA-seq data as presented by Poore et al. (18).

In this study, we proposed ASD-cancer (autoencoder-based subtypes detector for cancer), a semi-supervised deep learning framework based on autoencoder, a type of neural network that is trained to reconstruct its input data. It is composed of two components: an encoder that maps the input data to a lower-dimensional space, and a decoder that maps this representation back to the original data space. In our study, autoencoder models were used to extract relevant features from the normalized microbiome abundance data and transcriptome data for identifying cancer survival subtypes (Fig. 1b). These extracted features were then analyzed using univariate Cox-PH regression to identify a subset of survival-related features. To ensure an adequate number of features, we implemented an ensemble step using a total of 30 models. We then determine the number of survival subtypes using Gaussian Mixture Models and the highest silhouette score (Materials and Methods).

### Two survival subtypes detected in 20 types of cancers

Using ASD-cancer, we analyzed the host RNA-seq and tumor microbiome data (Materials and Methods) of 20 TCGA cancers. For every type of cancer, we determined the optimal clustering number K that yields the best silhouette score, a metric that measures clustering stability and accuracy. Our analysis revealed that setting K to 2 yielded the highest silhouette score for each of the 20 types of cancer, thereby enabling the detection of two distinct survival subtypes (Fig. 2a). Remarkably, the subtype with better prognosis outcomes was designated as ASD-1, while the subtype with worse prognosis outcomes was designated as ASD-2. Using log-rank tests, we identified statistically significant differences (log-rank test, $P < 0.05$) between the two subtypes' Kaplan-Meier curves. We also obtained high C-indexes (most are greater than 0.8, higher than the expected value of random models).

We also examined the distribution of age and clinical stage among the patients assigned to the two subtypes. Intriguingly, our analysis demonstrated that the age distributions of the two subtypes were similar across all types of cancer (Fig. S1). To investigate the relationship between the survival subtypes and clinical stages, chi-squared analyses were performed on the distribution of clinical stages between the two subtypes for each cancer. Our findings indicated that, among the 15 types of cancer with clinical stage information (Fig. S2a and b; Table S2), 9 of them, whose ASD-1 and ASD-2 could usually be differentiated clearly, exhibited significant differences (log-rank test, $P < 0.05$) in clinical stage distributions between the two subtypes. Among these seven types

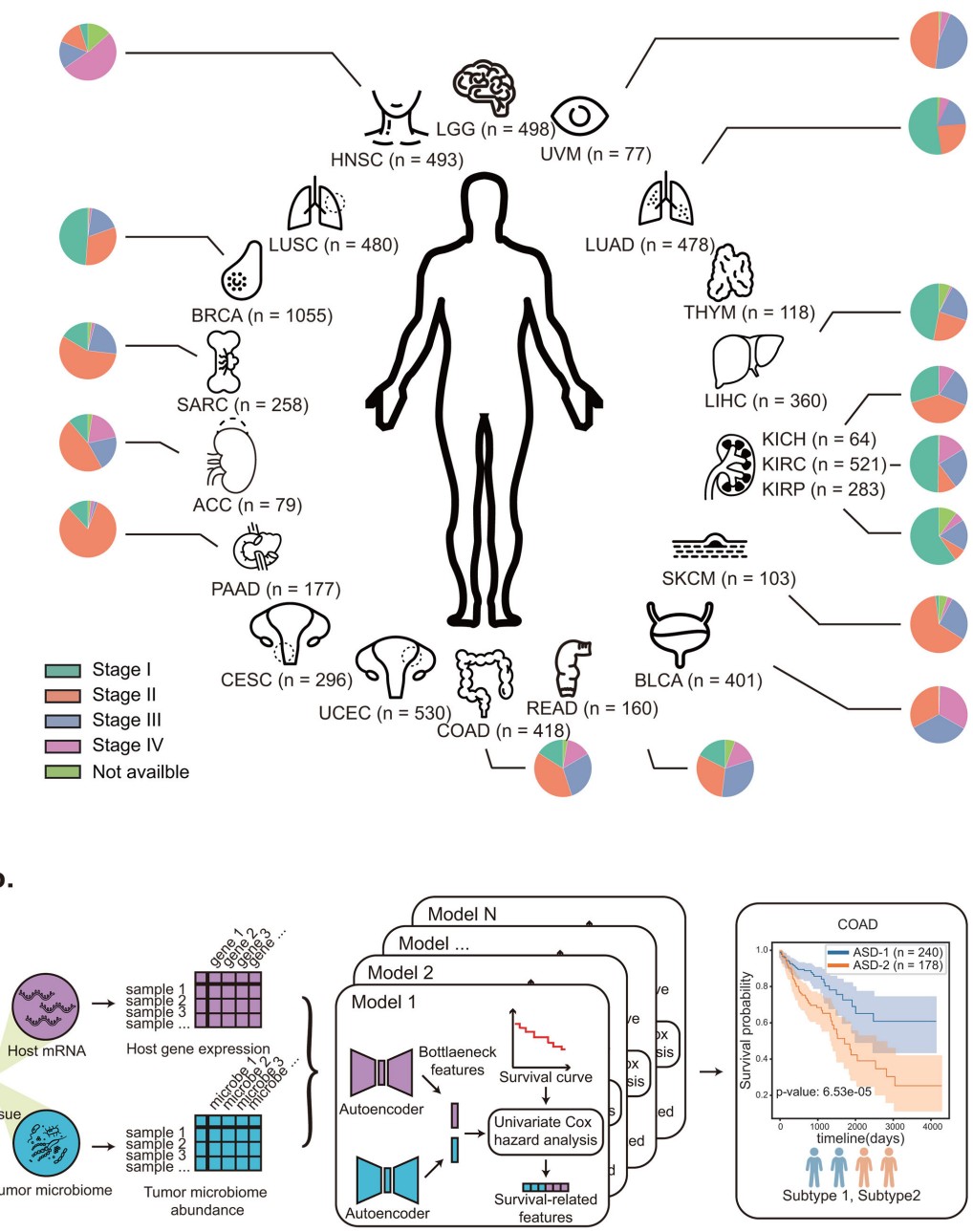

**FIG 1** Material and pipeline of survival subtypes detection. (a). We obtain the 20 cancer data sets from TCGA. Each data set contains paired RNA-seq data and tumor microbiome data. The pie plot near each cancer represents the distribution of tumor stages defined by AJCC Cancer Staging System. The abbreviated names of cancer: ACC, adrenocortical carcinoma; BLCA, bladder urothelial carcinoma; BRCA, breast invasive carcinoma; CESC, cervical squamous cell carcinoma and endocervical adenocarcinoma; COAD, colon adenocarcinoma; LUAD, lung adenocarcinoma; HNSC, head and neck squamous cell carcinoma; KICH, kidney chromophobe; KIRC, kidney renal clear cell carcinoma; KIRP, kidney renal papillary cell carcinoma; LGG, brain lower grade glioma; LIHC, liver hepatocellular carcinoma; LUSC, lung squamous cell carcinoma; PAAD, pancreatic adenocarcinoma; READ, rectum adenocarcinoma; SARC, sarcoma; SKCM, skin cutaneous melanoma; THYM, thymoma; UCEC, uterine corpus endometrial carcinoma; UVM, uveal melanoma. The number of samples followed the abbreviated names. (b). The pipeline of Ensemble deep learning-based survival subtype detection model.

of cancer, ACC and BLCA exhibited overall survival-related cancer stage distributions, indicating that the survival subtyping of these cancers is highly related to the clinical stage. Conversely, KIRC, KIRP, LIHC, PAAD, and COAD had a high proportion of a specific

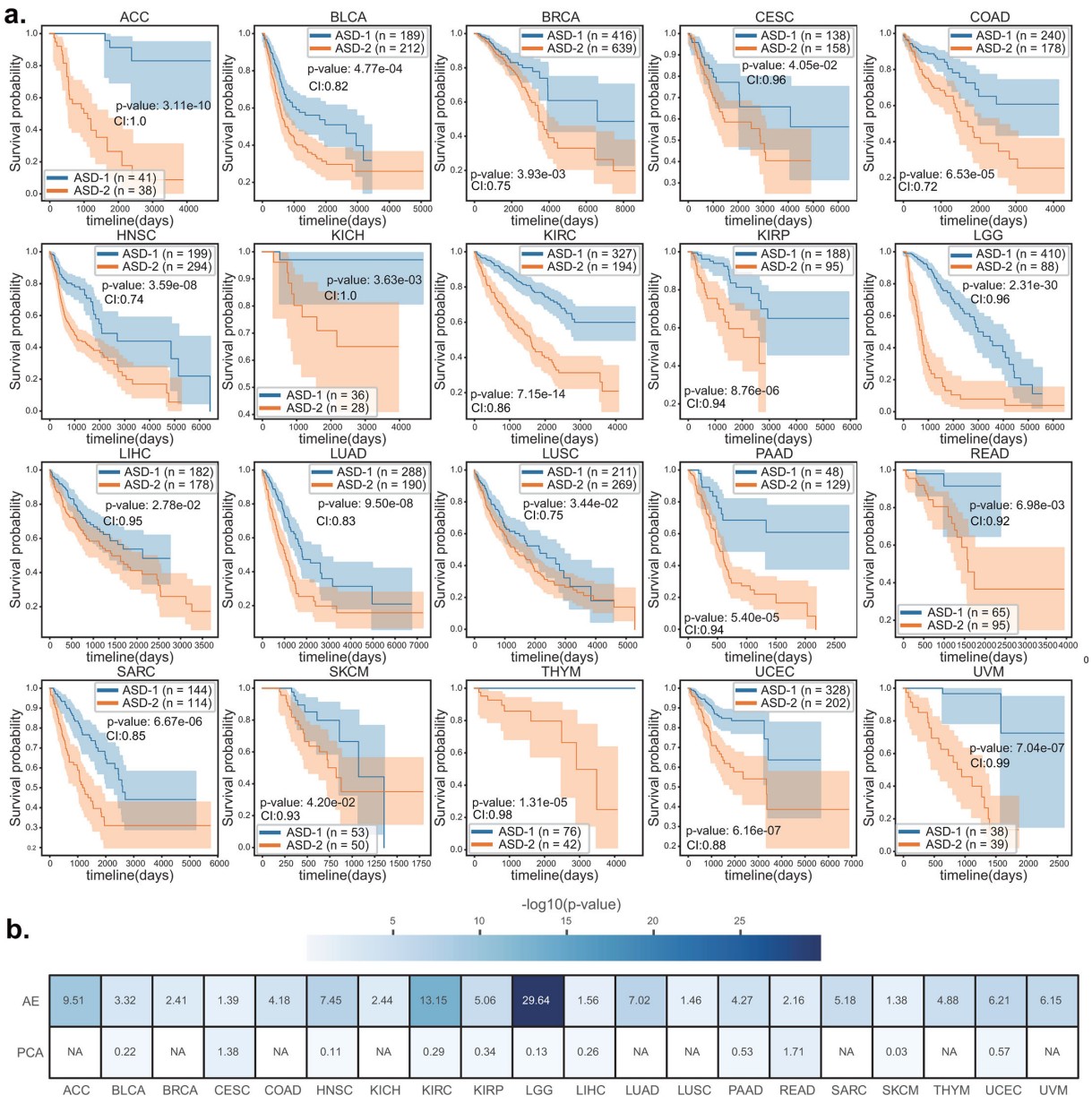

**FIG 2** Subtypes detection for the 20 TCGA cancer data sets. (a). Kaplan-Meier plots for each type of cancer. The survival curve for subtypes with better survival outcomes is marked in blue, while the curve for subtypes with worse survival outcomes is marked in orange. The *P* value is the result of a log-rank test, which is a statistical test used to compare the survival curves of different groups. The CI is concordance index, a measurement of how well a model predicts the ordering of patients' death times. (b). Heatmap showing the results of replacing the autoencoder with PCA. The values in the heatmap represent the −log10 of the *P* value from a log-rank test. "NA" indicates that no survival-related features were extracted from at least one of the two omics data sets (RNA-seq and tumor microbiome) for a type of cancer. AE: autoencoder.

cancer stage in one subtype, indicating that the prognosis of these cancers is highly related to a specific clinical stage. For instance, advanced-stage KIRC patients were more likely to be stratified into the low-risk group. In contrast, for the remaining six types of cancer whose ASD-1 and ASD-2 could usually not be differentiated clearly, we found no significant differences in the cancer stage distributions between the two subtypes. We propose that the prognostic outcome of these eight cancers may not be determined by clinical stages but by other molecular mechanisms.

## Enhanced survival subtyping through integration of multi-omics data

To ascertain the necessity of multi-omic data for survival subtyping, we exclusively employed features derived from single omic data to classify patients into distinct survival subtypes. Our analysis revealed that utilizing either the transcriptome or tumor microbiome alone yielded a lower statistically significant stratification (Fig. S3 and S4). For example, when integrated multi-omics data were utilized, the survival curves of ASD-1 and ASD-2 in HNSC patients exhibited significantly improved stratification (log-rank test, $P = 3.59E{-}8$) compared to the utilization of single omic data alone (log-rank test, $P = 5.13E{-}1$ for the transcriptome and $P = 1.84E{-}7$ for the tumor microbiome).

To further evaluate the performance, we conducted an additional benchmark analysis by replacing the autoencoder module in ASD-cancer with conventional PCA decomposition (Fig. 2b). Specifically, we transformed each omic data set into 100 new components and followed the same subsequent procedures as in ASD-cancer. The subtyping results indicated that PCA failed to extract survival-related features in nine cancers, and in 11 cancers, it demonstrated significantly inferior performance compared to the autoencoder module in ASD-cancer. We also compared ASD-cancer with Cox proportional hazards regression applied directly to the features. While Cox regression performed well, it failed to achieve significant stratification in certain cancer types (Fig. S5).

These findings highlighted the significance of integrating multi-omics data to achieve improved stratification and enhance our comprehension of the intricate mechanisms that influence cancer survival. Additionally, they demonstrate the potential and effectiveness of employing deep learning and ensemble learning techniques, as exemplified by the ASD-cancer model, to extract meaningful features that contribute to reliable cancer subtyping.

## Integrated multi-omics data shows high subtype prediction accuracy

We conducted an analysis of the alpha diversity of tumor microbiomes in two subtypes of different types of cancer. The results revealed noteworthy variations in tumor microbiome alpha diversity between the two subtypes of nine types of cancer (ACC, BRCA, CESC, HNSC, LIHC, LUSC, LGG, KICH, and KIRP) (Fig. S2c). Additionally, it was observed that the ASD-2 subtype exhibited elevated alpha diversity in their tumor microbiomes in comparison to the seven types of cancer (ACC, BRCA, HNSC, KICH, KIRP, LGG, and LIHC). We employed a random forest model to predict the survival subtypes of cancer (Fig. 3a; Fig. S6). We selected the top 20 most important features from each omics (Tables S2 and S3) and developed models based on these features. The outcomes obtained from this approach were either consistent with or superior to those obtained from using all features. Finally, the integration of the two omics was performed, and it was concluded that the model utilizing the 40 chosen features exhibited the highest level of precision. This model can be employed for prognosticating survival subtypes in forthcoming patients. We also benchmarked other machine learning models, including K-nearest neighbor, support vector machine, and Logistic Regression, with the random forest model demonstrating the best performance (Fig. S7).

We also employed two types of omics data to construct a random forest model for discriminating clinical stages among patients using the 40 features we selected above. Nonetheless, the accuracy attained levels for certain types of cancer were not deemed satisfactory (Fig. 3b). Specifically, the accuracy for PAAD was 0.86, however, this outcome could be attributed to the substantial proportion of stage I patients present in the sample. Ultimately, our attention was directed toward the differentiation of patients in stages I and IV for types of cancer in which stage I or IV samples represented more than 30% of the total. We also excluded SKCM from our analysis due to the small number of samples. The final model was applied to three types of cancer: ACC, COAD, and READ (Fig. 3c). The results indicate that the models for ACC and COAD performed well, and that gene expression was more informative for distinguishing between stages I and IV than

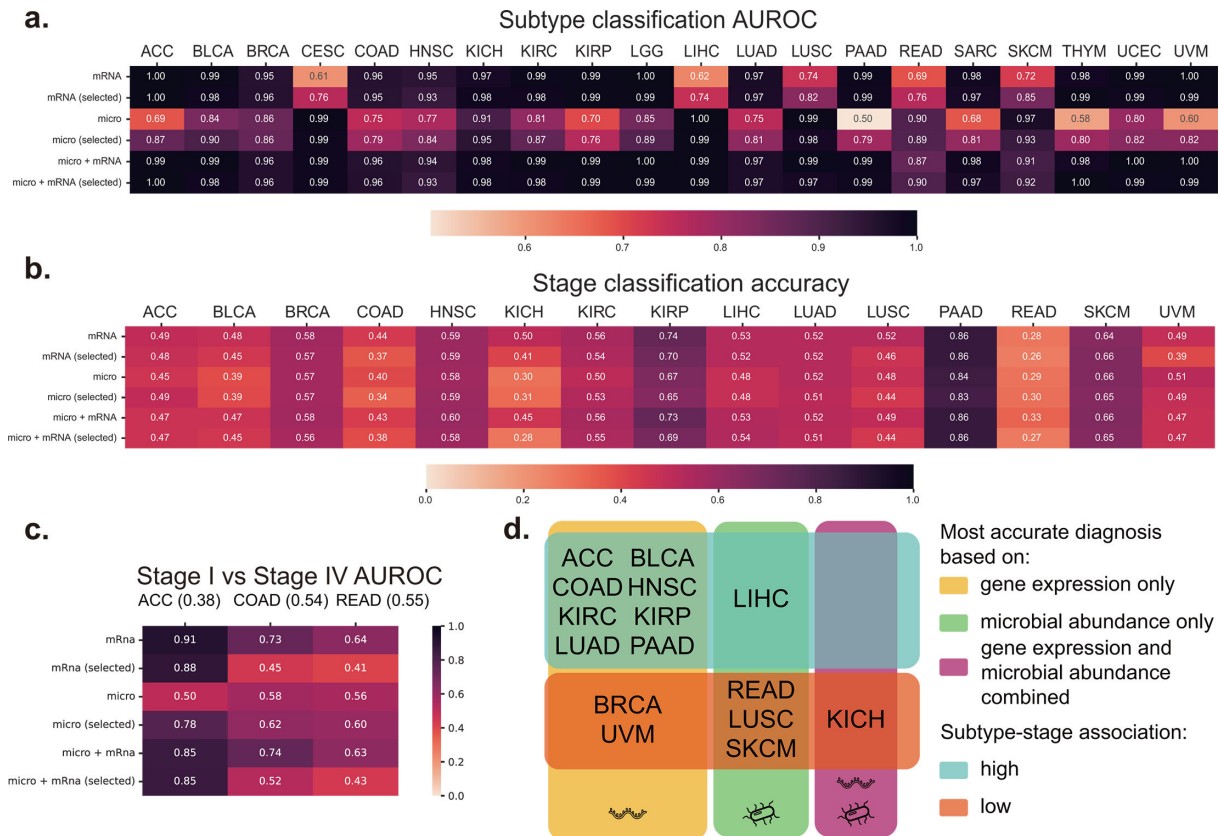

**FIG 3** Alpha diversity of tumor microbiomes in different survival subtypes and the prediction results of subtype and stage. (a) The area under receiver operating characteristic (AUROC) heatmap of the prediction results of survival subtypes using a leave-one-out method in random forest. Figure (b) shows the accuracy heatmap of the prediction results of tumor stages using a leave-one-out method in random forest, with the numbers after each type of cancer indicating the number of stages. (c) The accuracy heatmap of the prediction results of stages I and IV using a leave-one-out method in random forest, with the ratios after each type of cancer indicating the proportion of stage I samples among stages I and IV samples. In panels a–c, each row represents different features. The first row represents all microbiome features; the second row represents the top 20 most important microbiome features; the third row represents all transcriptome features; the fourth row represents the top 20 most important transcriptome features; the fifth row represents all features from both omics' approaches; and the sixth row represents the top 20 most important features from both omics' approaches. (d) The categories of cancers based on the results of omics predictions. The first category consists of cancers with better results obtained from transcriptomics data and the results are consistent with transcriptomics when the two omics are integrated. The second category consists of cancers with better results obtained from microbiome data and the results are consistent with microbiomes when the two omics are integrated. The third category consists of cancers with better results obtained from the integration of the two omics compared to using a single omics. Subtype-stage association: high means significant differences in clinical stage distributions between the two subtypes (chi-squared test, $P < 0.05$), while low means no significant differences in clinical stage distributions between the two subtypes (chi-squared test, $P > 0.05$).

the tumor microbiome. These findings suggest that biomarkers used to discriminate survival subtypes may not be sensitive to clinical stages.

Twenty types of cancer were classified into three distinct categories on the basis of the predictive capabilities of transcriptomic and microbiome data (Fig. 3d): best performance based on transcriptome only, microbiome only, and the integration of both omics. The first category exhibited superior transcriptome prediction results, the second category demonstrated better microbiome prediction results, and the third category showcased better outcomes from the integration of both omics. The results of our study indicate that the transcriptome is more strongly associated with predicting survival subtypes for the majority of cancers, with the exception of LIHC, READ, LUSC, and SKCM which exhibit a stronger correlation with the microbiome. The integration of both omics led to enhanced prediction accuracy for KICH, indicating a potential involvement of the interplay between tumor microbiomes and host genes in the survival of this cancer.

## Multi-omics features that play critical roles in the association of pre-defined cancer stages and intrinsic survival subtypes

In this study, we ascertain the molecular mechanisms that underlie the variations in survival outcomes observed among different cancer subtypes. We utilized gene set enrichment analysis (GSEA) to compare the association between pre-defined cancer stages and intrinsic survival subtypes. Our results indicate that the ASD-2 subtype exhibited a higher degree of enrichment in cancer-related pathways as compared to the ASD-1 subtype in cancers whose clinical stage is related to survival subtyping.

First, we analyzed BLCA, LIHC, and LUAD, whose survival-specific subtyping aligned with clinical stages. The result revealed that the ASD-2 subtype in BLCA demonstrated an increase in the MicroRNAs in Cancer and Apelin signaling pathways (Fig. 4a), which have a tumor-promoting effect via the PI3k/Akt pathway or the activation of Notch3 and STAT3 (29). We also observed that certain microbes such as *Flaviflexus and Candidatus Nitrosoarchaeum* may interact with host genes like CACNG7 and further modulate signaling pathways like MAPK in the ASD-2 subtype (30). Similarly, in LIHC (Fig. 4b), we found microbes such as *Isoptericola* also play a role in regulating host survival through interactions with host gene PDE1C, and regulation of purine metabolism pathways. The ASD-2 subtype showed enrichment in the HIF-1 and Sphingolipid signaling pathways, whose dysregulation is associated with tumor angiogenesis and cancer progression (31). In LUAD (Fig. 4c), we noted that microbes such as *Afipia* and *Centipeda* may interact with the host gene UGT1A1, further regulating the pentose and glucuronate interconversion pathways. This regulation supports cancer cell growth and survival by generating pentose phosphate for nucleic acid synthesis (32). We also found that for BRCA, and LUSC, whose survival-specific subtyping did not align with clinical stages, there were no significant correlations between the microbiome and host genes. The implication of this statement is that rather than the interaction between various omics, the distinct influence of omics primarily determines the prognosis of these cancers.

Finally, we analyzed cancers without clinical stage information, such as CESC (Fig. 4d). The ASD-2 subtype in CESC showed enrichment in pathways such as the insulin signaling pathway and GABAergic synapse pathway. The insulin signaling pathway can promote tumorigenesis by activating the TOR-eIF4E-S6K pathway and enhancing the insulin/PI3K signa (33), whereas GABAergic system can exert immunosuppressive effects by disrupting the functions of various peripheral immune cells (34). Furthermore, we found that certain microbes such as *Zunongwangia* and *Fermentimonas* may interact with host genes like PGR and regulate Chemical carcinogenesis pathways.

In summary, this work offered a deeper understanding of plausible omics patterns that may account for variations in survival results. The results of our study indicate that tumors exhibiting increased interaction between the microbiome of the tumor and the host gene are more strongly correlated with clinical stage and prognosis, whereas other tumors are more closely linked to single types of omics. Additional investigation is required to authenticate these discoveries and clarify the function of microbes in controlling host viability and pathways associated with cancer.

## Performance of ASD-cancer on external cohorts utilizing transfer learning

The ASD-cancer model has demonstrated its ability to predict the survival subtype of new individual samples using both tumor microbiome and gene expression features. However, it should be noted that the TCGA database primarily consists of patients from Western populations. In order to assess the generalizability of our model, we applied the same workflow to two non-western cohorts. By harnessing the unsupervised learning capabilities of autoencoders, ASD-cancer can effectively utilize the knowledge of pre-trained models to swiftly capture survival-related features in newly obtained small-scale data sets for survival subtyping. First, we utilized the Roelands et al. gathered AC-ICAM cohort, which included 246 COAD patients from Qatar (35). The results yielded a C-index of 0.89 and a log-rank *P* value of 1.59E−7 (Fig. 5a). Additionally, we developed a random forest model based on 40 features selected from random forest model based on the

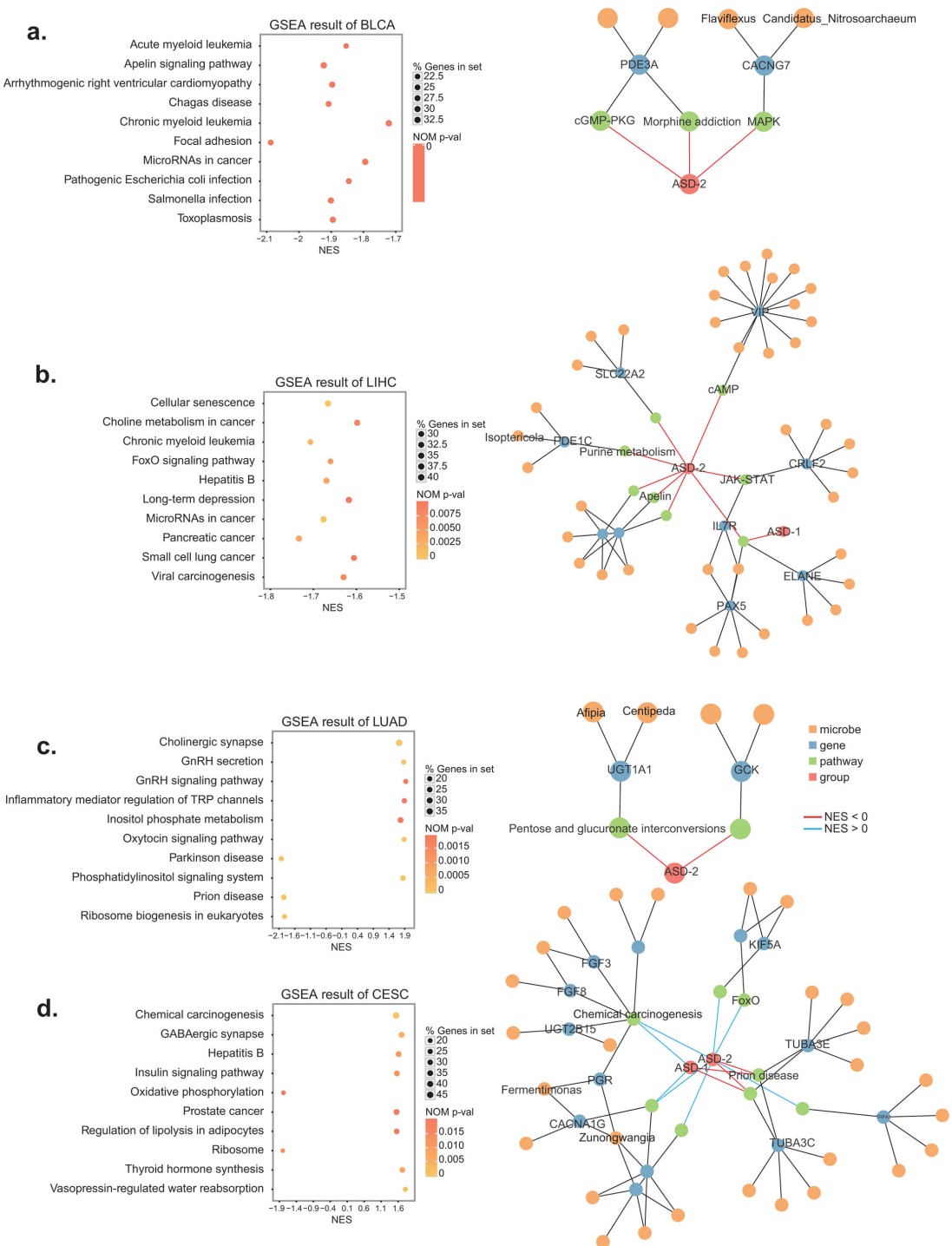

**FIG 4** GSEA result and correlation network for three representative cancers. (a) The top 10 enriched pathways for BLCA based on gene set enrichment analysis (GSEA) and the correlation network between microbes and host genes (for details see Materials and Methods). (b) The top 10 enriched pathways for LIHC based on GSEA and the correlation network between microbes and host genes (for details see Materials and Methods). (c) The top 10 enriched pathways for LUAD based on GSEA and the correlation network between microbes and host genes (for details see Materials and Methods). (d) The top 10 enriched pathways for CESC based on GSEA and the correlation network between microbes and host genes (for details see Materials and Methods). The depth of the point color represents the *P* value of enrichment, and the size of the point represents the number of genes enriched in the pathway. The value on the *x*-axis is the enrichment score, with positive values representing enrichment in subtypes with better survival, and negative values representing enrichment in subtypes with worse survival.

TCGA cohort and acquired an AUROC of 0.811 (Fig. 5b). Second, to further validate the patient survival risk stratification achieved by ASD-cancer, we conducted a test on another independent cancer data set obtained from 32 LIHC Chinese patients' tumor tissues collected by Huang et al. (36) The tumor microbiome abundance data were annotated according to Poore et al.'s pipeline (18). In this Chinese cohort, we successfully stratified the patients into ASD-1 and ASD-2 subtypes with a C-index of 0.98 and a log-rank $P$ value of 1.13E−2, utilizing our pre-trained models (Fig. 5c). The random forest model achieved a high AUROC of 0.877 based on 40 features selected from LIHC patients in the TCGA cohort (Fig. 5d).

In both validation cohorts, we emphasize the importance of the transfer learning strategy. Furthermore, the random forest model exhibited precise and efficient prediction without necessitating additional clinical variables or lower-dimensional feature transformations through the autoencoder approach. These findings support the reliability and diagnostic potential of the transfer learning strategy.

## DISCUSSIONS

Although the relationship between survival rates and clinical stage classification in cancer patients is debatable, it is generally acknowledged that, under ideal state, the survival outcomes get worse as the clinical stage progresses. However, patients with the same grade or stage demonstrate significant variability in outcomes, and more importantly, patients with different types of cancer have different survival-stage association patterns. Although several strategies for cancer survival subtyping based on molecular signatures have emerged, they typically only focus on a single omic (11) or the host itself (37), often disregarding other intricate factors within the TME, such as the tumor microbiome (13).

In this study, we identified two survival-specific subtypes, namely ASD-1 and ASD-2, by integrating data on the tumor microbiome and host gene expression. Notably, these subtypes do not entirely align with the clinical stage data. Seven types of cancer in our study showed high clinical stage and survival-subtyping associations, whereas eight types showed no difference in clinical stage distribution between the two survival subtypes (Fig. 6a). Our findings suggest that the tumor microbiome and host gene interact more actively in tumors with a high correlation between pre-defined cancer stages and intrinsic survival subtypes. Conversely, cancers with a low association between pre-defined cancer stages and intrinsic survival subtypes exhibit fewer interactivities between the tumor microbiome and host gene (Fig. 6b). These patterns indicate the two groups of cancer, depending on the interaction between host genes and tumor microbiomes: In the first group, host genes and tumor microbiomes have a weak correlation, and ASD-1 and ASD-2 can usually not be differentiated clearly. For example, ESCA in the first group has a log-rank test $P$ value of 3.40E−02. A more concrete example is LUSC, for which we found no strong correlation pathway between tumor microbiome and host gene ($R^2 < 0.9$), implying that the interactivity between tumor microbiome and host genes in these cancers is weak, but the prognosis of these cancers is primarily determined by the distinct influence of each omic, which is responsible for survival-specific subtyping in this particular cancer (log-rank test, $P = 3.25E−03$). In the second group, host genes and tumor microbiomes have a strong correlation, and ASD-1 and ASD-2 can usually be differentiated clearly. For example, LIHC in the second group has a log-rank test $P$ value of 8.12E−06.

Furthermore, we found that compared to the low-risk group, the high-risk group was more likely to have a greater number of pathways connected to cancer. This provides evidence that poor survival outcomes in cancer patients may be driven by distinct biological pathways. In addition, we found numerous possible microbial-gene interaction pathways that might contribute to cancer survival. For example, in the case of BLCA, *Flaviflexus and Candidatus Nitrosoarchaeum* may regulate host survival through interactions with host genes enriched in critical signaling pathways in cancer, particularly the MAPK signaling pathway. These species could be targeted for therapy to improve patient

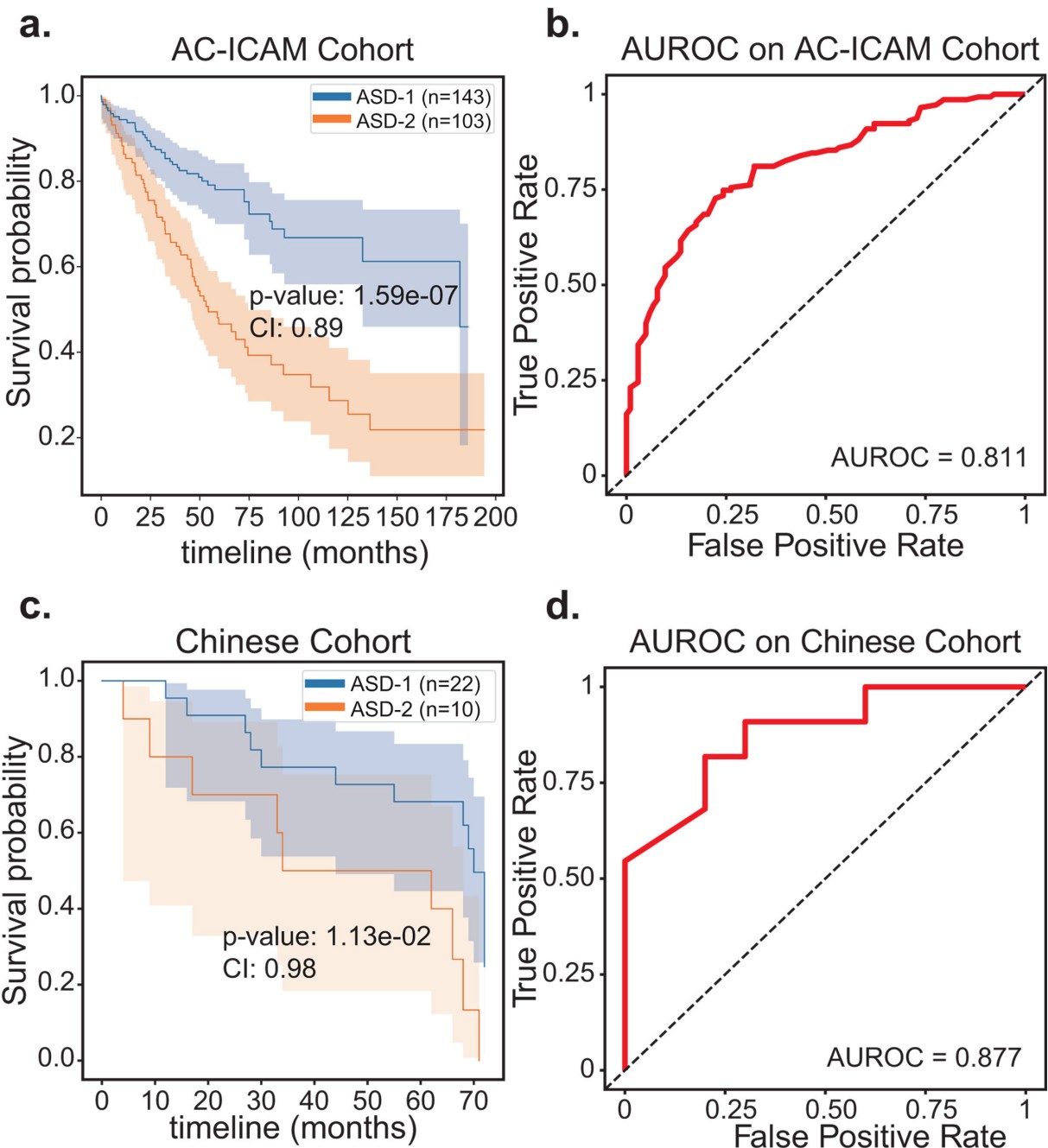

**FIG 5** Validation results on two external data sets. (a) Kaplan-Meier plots illustrating the survival subtypes of ASD-1 and ASD-2 in the AC-ICAM Cohort. (b) AUROC plot of the random forest model for predicting survival subtypes in the AC-ICAM cohort using leave-one-out validation. (c) Kaplan-Meier plots illustrating the survival subtypes of ASD-1 and ASD-2 in the Chinese cohort. (d) AUROC plot of the random forest model for predicting survival subtypes in the Chinese cohort using leave-one-out validation. The survival curve for subtypes with favorable survival outcomes is represented in blue, while the curve for subtypes with unfavorable survival outcomes is depicted in orange. The *P* value corresponds to the result of a log-rank test, a statistical test employed to compare survival curves across different groups. The CI refers to the concordance index, which quantifies the predictive ability of the model in ranking patients' time of death.

prognosis. This discovery is significant because it implies the tumor microbiome may play a pivotal role in determining cancer patients' chances of survival.

We also detected ASD-1 and ASD-2 on two external data sets and accurately predicted them using selected host genes and tumor microbes. Collectively, these patterns have not only demonstrated that microbes in the tumor microenvironment

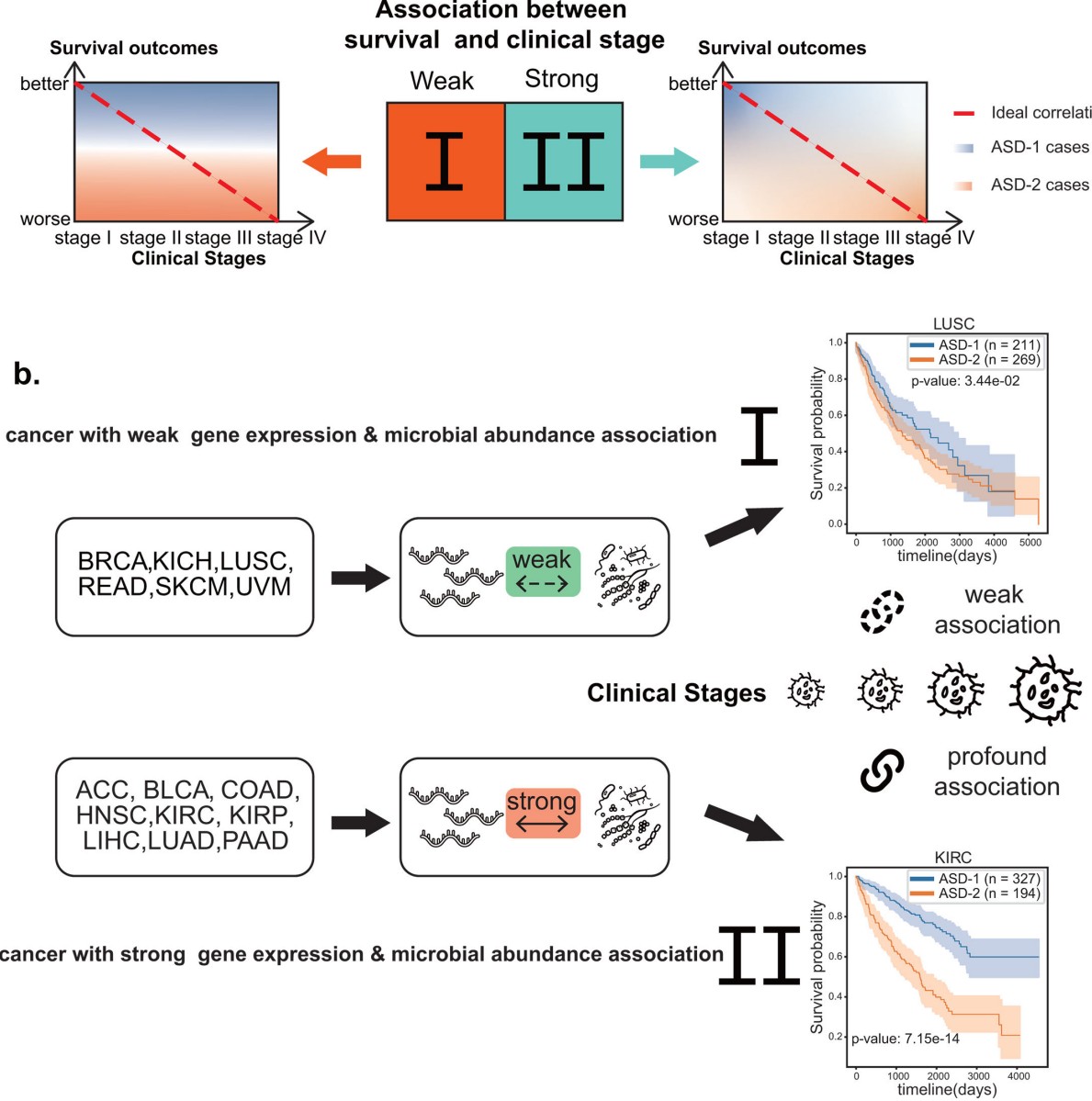

**FIG 6** Illustration of clinical stages of tumor and survival subtyping. (a) Two groups of cancer identified in our study with schematic heatmap illustrating the association between ASD-1 and ASD-2 with patients' survival outcomes across various clinical stages of tumors. The red dotted diagonal line represents the correlation between clinical stage and patients' survival outcomes under the ideal state. The *x*-axis denotes the clinical stages of the tumor, while the *y*-axis represents the survival outcomes. In the first group, there is weak association between clinical stages and survival subtypes, while in the second group, there is strong association between clinical stages and survival subtypes. (b) Schematic representation of the interaction between host genes and tumor microbiomes in the two groups of cancer. In the first group, host genes and tumor microbiomes are weakly correlated, and ASD-1 and ASD-2 usually indistinguishable; while in the second group, host genes and tumor microbiomes are strongly correlated, and ASD-1 and ASD-2 are typically distinguishable.

might influence cancer development in a variety of ways but also that we can stratify patients with high accuracy based on their prognosis using the tumor microbiome and host gene expressions, which is a critical addition to clinical stage information for monitoring cancer patients' status.

In conclusion, our study supports the idea that integrating transcriptome and microbiome data can aid in understanding the factors that influence cancer patients'

survival. Specifically, we focused on ASD-cancer among 20 types of cancer and discovered two survival subtypes, ASD-1 and ASD-2, which are not solely reliant on traditional clinical criteria such as cancer stage. Additionally, we successfully categorized 15 out of 20 types of cancer with clinical stage information into two groups, based on the relationship between pre-defined cancer stages and intrinsic survival subtypes, revealing different interactive patterns between host genes and the tumor microbiome. These findings have the potential to enhance the prognostic information accessible to clinicians and contribute to the developing field of precision medicine. Moreover, we validated the ASD-cancer strategy on two external small-scale cohorts, which highlighted our framework's transferability and value in clinical practice. Our study sheds light on the potential for using multi-omics data and deep learning methods to improve cancer prognosis and discover personalized therapeutic targets.

## MATERIALS AND METHODS

### TCGA data sets

Transcriptome data for this study was obtained from the TCGA database (https://brd.nci.nih.gov/brd/sop-compendium/show/701). Only SampleIDs containing "01A" were retained to ensure the data came from primary tumors. The tumor microbiome data were obtained from a previous cancer microbiome study (18), which included whole genome and whole transcriptome sequencing data of 33 types of cancers from the TCGA database. The data were processed using state-of-the-art tools to minimize sample contamination (38), and only data derived from whole transcriptome sequencing was used. The final result was a paired tumor microbiome and transcriptome data set comprising 20 types of cancer from tissue samples. A Strengthening The Organizing and Reporting of Microbiome Studies (STORMS) checklist (39) is available at https://doi.org/10.6084/m9.figshare.25152683.v1.

### ASD-cancer workflow

ASD-cancer workflow was designed in several modules as follows. The first module preprocesses the data, which involves normalization and scaling of both mRNA and microbiome data to ensure comparability between the two data sets. The second module transforms each omic feature into low-dimension representation using an autoencoder. An ensemble step was used to obtain adequate features by training 20 autoencoders for each cancer. The third module establishes cancer survival subtypes by performing a univariate Cox proportional hazards analysis and Gaussian mixture model clustering. The fourth module establishes the association of cancer survival subtypes and stages using chi-square tests and random forest models for subtype and stage prediction. The fifth module identifies the association between host gene expressions and microbial abundances using Pearson correlation analysis. Finally, gene enrichment analysis was performed to identify biological pathways associated with different cancer survival subtypes.

### Preprocessing

During the preprocessing stage, both mRNA and microbiome data were transformed to ensure compatibility for downstream analyses. For the mRNA data, we applied a rank-based normalization approach, where expression values were first ranked and then transformed to fit a standard normal distribution. This step helps reduce the influence of extreme values and ensures consistency across samples. For the microbiome data, we normalized the relative abundances by converting raw read counts into proportions, accounting for differences in sequencing depth across samples. After normalization, both data sets were scaled by subtracting the mean and dividing by the standard deviation for each feature. This standardization ensures that all features contribute equally, regardless of their original scales or units. By applying these preprocessing steps,

we addressed disparities in the magnitude and distribution of the mRNA and micro-biome data, facilitating reliable comparison and joint analysis during model development.

## Autoencoder transformation

Autoencoders are a type of neural network used for unsupervised learning that can be used to learn efficient data representations in an unsupervised way. The network consists of two main components: the encoder, which maps the input data to a lower-dimensional representation, and the decoder, which reconstructs the original data from the encoded representation. The general formula for an autoencoder can be written as follows:

$$X' = g(f(X))$$

where $X$ is the input data, $f$ is the encoder function, $g$ is the decoder function, and $X'$ is the reconstructed output.

In this study, we utilized an autoencoder to transform each omic data into a lower-dimensional representation. We implemented an ensemble step to obtain adequate features by training 30 autoencoders for each cancer. The number of autoencoders was determined through a gradient experiment, varying from 5 to 40 and assessing clustering consistency at each step (Fig. S8). Each autoencoder had an encoder with two linear layers of size 500 and 100, respectively, with a dropout rate of 0.2 and Tanh activation function in between. The decoder also had two linear layers of size 500 and the length of each omic feature, respectively, with a dropout of 0.2 and Tanh activation function in between. The optimal hidden layer size for each model was determined via grid search (Fig. S9). We applied early stopping based on a validation split to prevent overfitting. Specifically, we reserved 20% of the training data as a validation set and monitored the validation loss during training. Training was halted once the validation loss failed to decrease over five consecutive epochs, ensuring that the model was not overfitted to the training data. In the end, we obtained 100*30 lower-dimensional features for each cancer.

## Cancer survival subtyping

Cancer survival subtyping was established by performing a univariate Cox proportional hazards (Cox-PH) analysis on 100*30 features that had been reduced in dimension using autoencoder, along with survival information for the samples. We identified the features that were significantly associated with survival by selecting those with $P$ values < 0.05. Using all the selected features, we performed Gaussian mixture model (GMM) clustering with $K$ values ranging from 2 to 5. We compared the silhouette score for each cluster, and selected the $K$ value that resulted in the highest silhouette score as the number of clusters. Once we obtained the clustering results, we plotted Kaplan-Meier curves for each cluster and performed the log-rank test to assess the differences in survival between the clusters. We used the lifelines package (40) for Cox-PH analysis, Kaplan-Meier curve plotting, log-rank test, and the scikit-learn package (41) for GMM clustering.

## Establishment of the association of cancer survival subtypes and stages

We first analyzed the distribution of clinical stage across different cancer subtypes and used a chi-square test to evaluate whether there were significant differences in clinical stage distribution between the subtypes. Next, we used a random forest model to predict the cancer subtype and clinical stage for each sample. For subtype prediction and clinical stage prediction, we used leave-one-out cross-validation to calculate the AUROC and selected the top 20 features based on their importance scores. We selected cancers for stages I and IV prediction based on the proportion of samples in stage I or IV

was at least 30% and excluded skin cutaneous melanoma (SKCM) due to a small sample size.

## Establishment of the association of host gene expressions and microbial abundances

To explore the relationship between host gene expressions and microbial abundances, we first calculated the correlation coefficients between each host gene and microbial species using Pearson correlation analysis. Specifically, we utilized PyTorch to implement the correlation analysis and utilized GPU acceleration to speed up the computation. We only selected microbiome-host gene pairs with a correlation coefficient greater than 0.9 to ensure high confidence in the correlation.

To identify the biological pathways associated with different cancer survival subtypes, we performed gene enrichment analysis using the gseapy package (42). This package provides access to various databases and tools for gene set enrichment analysis, including the KEGG and Reactome databases. We calculated the enrichment scores for each pathway and used the $q$-value to select the top 10 enriched pathways for visualization.

## Transfer learning strategy

In this study, transfer learning is employed to enhance the generalizability of our ASD-cancer model across different populations. Specifically, the transfer learning strategy is achieved by initializing the model with weights pre-trained on TCGA data set, which primarily consists of Western population data. This enables the model to leverage the learned representations from the large-scale TCGA cohort, thus capturing important survival-related features from tumor microbiome and gene expression data. By doing so, the model can efficiently adapt to new, smaller data sets, even when the population demographics differ.

To test this generalizability, we applied the pre-trained ASD-cancer model to two independent, non-Western cohorts: the AC-ICAM cohort of 246 COAD patients from Qatar, and a cohort of 32 LIHC patients from China. In both cases, the pre-trained weights allowed the model to quickly capture relevant features and perform accurate survival subtyping. The use of the pre-trained model in these cohorts significantly improved performance, as demonstrated by the high C-index and AUROC scores.

## Validation of data sets and analyses

For the AC-ICAM cohort, we utilized a cohort of 246 COAD patients collected by Roelands et al. (35) Clinical information and gene expression data were obtained through cBioportal, while microbiome abundance data were acquired from FigShare.

For the Chinese cohort, we utilized 32 liver cancer tissue samples collected by Huang et al. (36). We downloaded the raw RNA-seq sequencing data (PRJNA576155) from the Sequence Read Archive (SRA) and performed downstream analysis to obtain gene expression levels as well as tumor microbiome abundance data.

## Gene expression data

To obtain gene expression data, we employed HISAT2 (43) for RNA-seq alignment, using hg19 as the reference genome. We utilized SAM-tools (44) for data conversion and sorting of the alignment results. Finally, we quantified gene expression using StringTie (45), selecting hg19 as the reference transcriptome, and extracting the FPKM values as the expression levels.

## Tumor microbiome data

To obtain tumor microbial abundance data, we employed Kraken2 to perform species annotation on sequences that could not be aligned to the human genome (46). We

utilized the standard Kraken2 database (built in 2023.6.1), which includes both bacteria and viruses. After annotation, we selected only the microbial taxa that matched the annotation results obtained by Poore et al. (18). Finally, we computed the relative abundances of these selected microbial taxa.

## Computational environment

All experiments were conducted in a high-performance computing environment. The system was equipped with an Intel(R) Xeon(R) CPU E5-2630 v4 @ 2.20 GHz, with 256 GB of memory, and a Tesla K80 GPU for accelerated computations. The operating system used was CentOS Linux release 7.9.2009. The Autoencoder models were implemented using PyTorch library and executed in this environment to ensure efficient processing of large-scale data sets.

## Visualization of results

We utilized several software tools to visualize the results of our analyses. Cytoscape was used to create the microbiome-host gene correlation network. Kaplan-Meier curves were plotted using the lifelines package to compare the survival of different subtypes. For other types of visualizations such as AUROC and accuracy heatmaps, we utilized the plotnine and seaborn packages.

## Statistical analysis

All statistical analyses were performed using Python packages, including Scikit-learn, scipy, gseapy, and lifelines. The significance level was set at 0.05, and all $P$ values were two-sided.

## ACKNOWLEDGMENTS

This work was partially supported by the National Natural Science Foundation of China (grant nos. 32071465 and 31871334) and the National Key R&D Program of China (grant nos. 2023YFA1800900 and 2018YFC0910502). Numerical computations were performed at the Hefei Advanced Computing Center.

H.Z. and K.N. conceived and proposed the idea and designed and developed the framework. H.Z. and X.X. performed the experiments and analyzed the data. H.Z., X.X., and M.C. visualized the data. L.J. provided valuable data. H.Z., X.X., M.C., L.J., and K.N. contributed to editing and proofreading the manuscript. All authors read and approved the final manuscript.

## AUTHOR AFFILIATIONS

[1]Key Laboratory of Molecular Biophysics of the Ministry of Education, Hubei Key Laboratory of Bioinformatics and Molecular-Imaging, Center of AI Biology, Department of Bioinformatics and Systems Biology, College of Life Science and Technology, Huazhong University of Science and Technology, Wuhan, Hubei, China
[2]Geneis Beijing Co., Ltd., Beijing, China
[3]Qingdao Geneis Institute of Big Data Mining and Precision Medicine, Qingdao, China

## AUTHOR ORCIDs

Haohong Zhang  http://orcid.org/0000-0001-6267-4244
Mingyue Cheng  http://orcid.org/0000-0003-1243-5039
Kang Ning  http://orcid.org/0000-0003-3325-5387

## FUNDING

| Funder | Grant(s) | Author(s) |
|---|---|---|
| MOST \| National Natural Science Foundation of China (NSFC) | 32071465,31871334 | Kang Ning |
| MOST \| National Key Research and Development Program of China (NKPs) | 2018YFC0910502 | Kang Ning |

## AUTHOR CONTRIBUTIONS

Haohong Zhang, Conceptualization, Data curation, Formal analysis, Methodology, Software, Validation, Visualization, Writing – original draft, Writing – review and editing | Xinghao Xiong, Methodology, Visualization, Writing – review and editing | Mingyue Cheng, Writing – original draft | Lei Ji, Resources | Kang Ning, Conceptualization, Funding acquisition, Methodology, Supervision, Writing – original draft, Writing – review and editing

## DATA AVAILABILITY

Transcriptome data for this study were obtained from the TCGA database (https://brd.nci.nih.gov/brd/sop-compendium/show/701). The tumor microbiome data were obtained at ftp://ftp.microbio.me/pub/cancer_microbiome_analysis/. All source code has been uploaded to https://github.com/HUST-NingKang-Lab/ASD-cancer.

## ADDITIONAL FILES

The following material is available online.

### Supplemental Material

**Supplemental material (mSystems01395-24-s0001.docx).** Supplemental figures and tables.

### Open Peer Review

**PEER REVIEW HISTORY (review-history.pdf).** An accounting of the reviewer comments and feedback.

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
