## [Reviewer comments · mSystems]

Deep learning enabled integration of tumor microenvironment microbial profiles and host gene expressions for interpretable survival subtyping in diverse types of cancers

Haohong Zhang, Xinghao Xiong, Mingyue Cheng, Lei Ji, and Kang Ning

Corresponding Author(s): Kang Ning, Huazhong University of Science and Technology

Review Timeline:

Submission Date:

October 18, 2024

Accepted:

October 22, 2024

Editor: Jack Gilbert

Reviewer(s): The reviewers have opted to remain anonymous.

Transaction Report:

DOI: <https://doi.org/10.1128/msystems.01395-24>

Re: mSystems01395-24 (Deep learning enabled integration of tumor microenvironment microbial profiles and host gene expressions for interpretable survival subtyping in diverse types of cancers)

Dear Prof. Kang Ning:

Your manuscript has been accepted, and I am forwarding it to the ASM production staff for publication. Your paper will first be checked to make sure all elements meet the technical requirements. ASM staff will contact you if anything needs to be revised before copyediting and production can begin. Otherwise, you will be notified when your proofs are ready to be viewed.

Sincerely,
Jack Gilbert
Editor
mSystems